# Laboratory assessment of alternative stream velocity measurement methods

**Stephen Hundt**⦿*, **Kyle Blasch**⦿

Idaho Water Science Center, US Geological Survey, Boise, Idaho, United States of America

* shundt@usgs.gov

**Data Availability Statement:** All data produced for this manuscript can be accessed through ScienceBase at: https://doi.org/10.5066/P9XDFP4M. Hundt, S.A., 2019, Data from a Laboratory Assessment of Alternative Stream

## Abstract

Understanding streamflow in montane watersheds on regional scales is often incomplete due to a lack of data for small-order streams that link precipitation and snowmelt processes to main stem discharge. This data deficiency is attributed to the prohibitive cost of conventional streamflow measurement methods and the remote location of many small streams. Expedient and low-cost streamflow measurement methods used by resource professionals or citizen scientists can provide scientifically useful solutions to this data deficiency. To this end, four current velocity measurement methods were evaluated in a laboratory flume: the surface float, rising body, velocity head rod, and rising air bubble methods. The methods were tested under a range of stream velocities, cross-sectional depths, and streambed substrates. The resulting measurements provide estimates of precision and bias of each method, as well as method-specific insight and calibration formulas. The mean values of the coefficient of variation, a measure of precision, were 10% for the surface float, 10% for the velocity head rod, 14% for the rising body, and 9% for the air bubble method. The values of scaled mean error, a measure of bias, were -8% for the surface float, -4% for the velocity head rod, -1% for the rising body, and 4% for the air bubble. The velocity head rod and surface float methods were the easiest methods to use, providing greater precision at large (> = 0.6 m/s) and small (<0.6 m/s) velocities, respectively. However, the reliance on a velocity ratio for each of these methods can generate inaccuracy in their results. The rising body method is more challenging to execute and of lower precision than the former two methods but provides low bias measurements. The rising air bubble method has a complex instrument assembly that is considered impractical for potential field user groups.

## Introduction

Streamgaging networks provide a fundamental and necessary dataset for water-resources and environmental management. Public agencies, non-profit organizations, academic institutions, and others collect continuous, high-precision streamflow data using fixed stream gauge networks. The cost of maintaining these networks limits their placement to easily accessible streams and prohibits the gathering of datasets with high spatial density or widespread coverage [1,2]. The major expenses of a streamgage network—the specialized labor force, significant

Velocity Measurement Methods: U.S. Geological Survey data release, https://doi.org/10.5066/P9XDFP4M.

**Funding:** The Northwest Climate Adaptation Science Center and the United States Geological Survey funded the design and execution of this study.

**Competing interests:** The authors have declared that no competing interests exist.

travel times to gage locations, and the purchase and maintenance of specialized equipment–are difficult to avoid and leave few opportunities to reduce costs. This leaves data lacking for the type of watershed-wide understanding and modeling that is often important for informing land-management and policy decisions [2–4]. Species vulnerability assessments (aquatic and terrestrial), land management actions (e.g., forest management, regulations on aerial application of pesticides and fire retardants), and water-quality regulation often rely upon understanding hydrologic characteristics of many streams throughout a watershed [5–8].

An opportunity exists to enlarge streamflow datasets by introducing simple, low-cost streamflow measurement methods that can be used by resource managers, citizen scientists, and others. By reducing the cost of purchasing and maintaining specialized equipment and by adding streamflow measurements to planned trips, the density and coverage of streamflow measurements may be expanded beyond what is currently feasible with traditional streamgaging networks. Expedient, low-cost methods for measuring streamflow in small streams using the velocity-area method could be used to gain an understanding of streamflow characteristics, such as hydrograph shape and the timing and magnitude of peak and low flows, and does not necessarily require using continuous, high-precision stream gauges. Such methods have been previously explored in field settings [9], yet the methods have not been fully studied in laboratory settings to provide a basic understanding of the origination of measurement errors.

For streamflow measurements to be used for management or research, the methods must have minimal acceptable performance metrics (precision and bias) and the ability to standardize instrumentation and methodology to provide consistency in repeat measurements. The objective of this study was to evaluate the potential of four current velocity measurement methods that may be used with the velocity-area procedure to quantify stream discharge. These methods were tested in an experimental flume to determine their ability to measure velocity under varying flow and substrate conditions, as well as to evaluate the ability to standardize the equipment and make repeat measurements. The four stream velocity measurement methods selected include the surface float, velocity head rod [10] [11], rising body, and rising air bubble methods [12] [13].

## Materials and methods

### Traditional streamflow measurements

Each of the four stream velocity methods can be used to compute a streamflow measurement by using the velocity-area procedure. Under this procedure, discharge is calculated by integrating the product of the current velocity and the cross-sectional area of the stream [14]. In a wading measurement, the stream width is divided into multiple rectangular/trapezoidal shaped "verticals" or subsections at which a depth, width, and velocity measurement are made. The sectional depth, width, and velocity are multiplied to compute a sectional discharge, and the products of the sections are then summed to compute a full stream discharge [14]. Computation is accomplished with the formula $Q = \Sigma(A^*V)$, where Q is the total discharge, V is the mean velocity of a subsection, and A is the area of a subsection.

This approach is accurate so long as each measurement is made at a depth near the average velocity within each subsection. As depth and velocity both vary laterally across a stream, one strategy for improving accuracy is to divide the stream into multiple verticals. Velocity also varies with stream depth and collecting a good approximation of the depth-averaged velocity typically relies upon an assumption of how velocity varies with depth. Streams tend to display a log-parabolic velocity profile, with velocities smallest at the frictional bed of the stream and increasing towards a maximum velocity just below the surface [15]. Under this velocity profile, a single measurement taken at 0.6 of the stream depth provides a good approximation of the

depth-averaged velocity. Under many conditions, such as in streams with high relative bed roughness, stream velocities do not display a log-parabolic profile. In these cases, a two-point method is used in which measurements are taken at both 0.2 and 0.8 of the stream depth. The two-point method is also commonly employed for its better accuracy under a wider range of stream conditions [14].

Low-cost alternatives to the types of current meters used in a professional setting are unable to collect a velocity at a specific depth. The methods tested in this study instead measure a depth-averaged or surface velocity. The surface float and velocity head rod methods share a challenge present in other emerging technologies such as large-scale particle image velocimetry (LSPIV) [16] and surface velocity radar (SVR) [17] that measure surface velocities. To compute a discharge using these methods, a relationship must be assumed between the surface and depth-averaged velocity. This relationship, called the velocity ratio, is defined as the ratio of depth-averaged velocity to the surface velocity. Recent effort has been put towards measuring the velocity ratio under different conditions. It has been observed [18] that the velocity ratio tends towards a value of 0.85 for deeper streams with smoother substrates, but that greater variability is present for shallower streams with rougher substrates. Recognizing that the choice of a velocity ratio value may be a significant source of error and bias for surface methods in the small streams that we are targeting, two methods, the rising body and the rising air bubble methods, were also included in this study. These methods sample the entire stream depth and bypass the need to assume anything about the velocity profile of the stream. All four methods are described in greater detail below.

## Velocity measurement methods

**Surface float.**   The surface float method measures the time required for a buoyant object to float along the surface of a stream over a given distance. If the object is only partially submerged, and there is no wind, the object's velocity should equal the water velocity at the water surface. For the surface float method, a water-fillable plastic fishing float (Tough Bubble) was used as the floating object and a floating fishing line was strung through the float for easy retrieval. Weights were tied to a 1.26-meter hiking pole that was placed along the bottom of the flume parallel to the direction of flow and that served as a reference distance.

A measurement was made by dropping the float at the upstream end of the reference distance and using a stopwatch to record the time it took for the float to travel to the downstream end of the reference distance. The surface velocity was calculated by dividing the reference distance by the time of passage.

**Velocity head rod.**   The velocity head rod method takes advantage of the water buildup, or head, that develops when an object impedes the path of flowing water. The velocity of the free-flowing water and the shape of the rod drive the height of this head. The velocity head rod method measures the integrated velocity over a portion of the stream depth, though it remains a question whether the full depth-averaged velocity or the average velocity of some shallower portion of the water column is measured.

For the velocity head rod method, a 1 1/8" x 1 1/8" x 4' clear plastic corner guard (Trimaco®) was used as a velocity head rod and a separate, clear 12-inch ruler was used to measure the height of water buildup. While not as wide as other velocity head rods described in the literature, the corner guard has a few advantages. It is a commercial product with a standardized shape, allowing others to acquire the exact same product or another product of the same dimensions and to thus apply the calibrated velocity formula developed in the laboratory. Other advantages of the corner guard are that is transparent to allow for easier viewing of the

head buildup, it is lightweight for easy transport, and that it has a small surface area and bracketed shape to reduce deformation caused by fast currents.

To make a measurement, the velocity head rod is placed vertically into the stream with one end placed on the streambed and the other end above the water surface. The rod is oriented so that the open 90-degree angle of its bracketed length is facing upstream, with the user standing downstream of the rod. At sufficient velocity (about 15 cm/sec), water will build up on the upstream side of the rod. The difference in water level, or head, between the upstream and downstream side is visible through the rod and measured using the clear ruler. It is assumed that the kinetic energy of the free-flowing water is transferred entirely to potential energy when it is stopped behind the rod. This results in the following formula to translate the head difference into a velocity:

$$V_u = \sqrt{2gh} \qquad (1)$$

where, $V_u$ is uncalibrated velocity, $g$ is gravitational acceleration, and $h$ is the head difference.

A calibration formula derived from the results of the laboratory experiment is then applied to convert the uncalibrated velocity to the final depth-averaged velocity.

**Rising body.** To compute a water velocity, the rising body method measures: (1) the time it takes for an object to rise from the streambed to the surface; (2) the distance traveled downstream as the object rises. In theory, the distance traveled by the object is controlled by the entire velocity profile, which should allow the method to directly measure a depth-averaged velocity. We used a rigid, water-fillable plastic fishing float (Tough Bubble) as the rising body and a floating yardstick to measure downstream distance. The buoyancy of the float, and thus the downstream distance it traveled, could be altered by filling the float with varying volumes of water.

A measurement with the rising body method is made by releasing the float from the bottom of the stream and measuring the elapsed time and the distance traveled downstream as it rises to the surface. The depth-averaged velocity is calculated as the distance traveled divided by the elapsed time. The buoyancy of the float can be adjusted by filling it with water, which allows the user to control the time and downstream distance traveled. During the laboratory tests, the volume of water inside each float was measured and recorded. While the method is not as simple to perform as other methods, it has the advantage that the depth-averaged velocity is measured directly without the need to find an appropriate velocity ratio or calibration formula.

**Rising air bubble.** Like the rising body method, the rising air bubble method measures the distance an air bubble travels downstream as it rises from the streambed to the surface. This too, in theory, should provide a direct measurement of the depth-averaged water velocity. The rising air bubble method was selected for its potential ease of use and theoretical accuracy. The bubbling device was constructed using an aquarium pump, 0.635 cm (¼ inch) vinyl tubing, two pressure compensating drip irrigation nozzles, a semi-flexible drip irrigation sprinkler head, and a hiking pole. The pump, hoses, nozzles, and sprinkler head were affixed to the hiking pole and oriented with the sprinkler head parallel to the stream bottom and perpendicular to the direction of stream flow. Using this approach, consistently sized bubbles of about ½ cm in diameter were emitted from the sprinkler head.

A measurement with the rising air bubble method is taken by recording the downstream distance traveled by the stream of air bubbles as they rise from the bottom of the stream to the water surface. The depth-averaged velocity (V) is equal to the downstream distance ($d_d$) travelled by a bubble divided by the time (t) taken for it to rise to the surface:

$$V = d_d/t \qquad (2)$$

Rather than timing the air bubble, it is assumed that the vertical velocity ($V_v$) remains constant with depth. This value, which is derived in the laboratory, is equal to:

$$V_v = d/t \qquad (3)$$

Where, $d$ is equal to the water depth. Rearranging Eq (3) to solve for $t$ and substituting it into Eq (2) yields the final formula use to convert the measured downstream distance to a a depth-averaged water:

$$V = d_d * V_v/d \qquad (4)$$

Where, $V$ is the depth-averaged water velocity (assumed constant), $d_d$ is the downstream distance traveled by the float, $V_v$ is the vertical velocity of the air bubble, and $d$ is the water depth.

**Flume experiment.** The four methods were tested in the University of Idaho's high-gradient hydraulic flume. This flume is 20 m long, 2 m wide, 1.2 m deep, can produce a maximum discharge of 1.1 m³/s, and can be adjusted to a maximum slope of 10 percent. Measurements were made for a range of flume conditions, with water depths ranging from 10 cm to 75 cm, depth-averaged velocities ranging from 9 cm/s to 117 cm/s, and three different substrates: the smooth metal flume bottom, a gravel bed (clast diameter 2–5 cm), and a cobble bed (clast diameter 5–15 cm). The full set of 26 flume conditions are shown in Table 1. These were chosen to capture a range of safely wadable conditions that are typically encountered in small basin and montane streams.

Multiple vertical velocity profiles were measured for each set of conditions before, during, and after the four methods were tested. An acoustic Doppler velocimeter (ADV) was used to measure velocities at 1- to 5-centimeter depth intervals through the water column, and a particle image velocimeter (PIV) was used to measure velocity at the water surface. The velocities measured with the four alternative methods were compared to velocities measured with the ADV (water column) and PIV (surface) to determine the performance metrics of the four methods. The ADV and PIV measurements, while subject to error themselves, are regarded as "truth" in this analysis.

For each set of flow conditions, multiple velocity measurements were made using each of the four methods. Fifteen repeat measurements were made with each of the rising body, surface float, and rising air bubble methods. Ten repeat measurements were made with the velocity head rod method. A procedure was developed for collecting repeat measurements with the rising air bubble and velocity head rod methods to prevent establishing an initial bias about the measurement duration that would influence remaining measurements and result in underestimates of measurement variance. For the rising air bubble method, the yardstick was adjusted after each measurement so that the nozzle was located below a different position on the yardstick every time, with the starting and ending distance recorded for each measurement to calculate the total distance traveled by the air bubbles. For the velocity head rod method, each of five data collectors measured the head buildup twice: once in inches and once in centimeters. This resulted in ten repeat measurements.

## Data analysis

**Velocity profiles.** The multiple ADV and PIV profiles that were collected before, during, and after each set of measurements were averaged for each set of conditions to create a single representative velocity profile for that set of conditions. For six of the 26 conditions, the PIV did not record a usable surface velocity. In these cases, a surface-velocity value was instead extrapolated by fitting a power law curve to the ADV measurements. A velocity of zero was assumed for the flume bottom where the ADV was unable to measure. This is a commonly

**Table 1. Flume test conditions.**

| Measurement Date | Substrate | Water Depth (cm) | Depth-Averaged Velocity (cm/s) |
|---|---|---|---|
| 11/15/2016 | Gravel | 40 | 9 |
| 11/15/2016 | Gravel | 75 | 10 |
| 11/15/2016 | Gravel | 20 | 37 |
| 11/15/2016 | Gravel | 38 | 48 |
| 11/15/2016 | Gravel | 24 | 63 |
| 11/15/2016 | Gravel | 40 | 64 |
| 11/17/2016 | Cobble | 40 | 9 |
| 11/17/2016 | Cobble | 20 | 21 |
| 11/17/2016 | Cobble | 20 | 40 |
| 11/17/2016 | Cobble | 40 | 55 |
| 11/17/2016 | Cobble | 22 | 78 |
| 11/17/2016 | Cobble | 38 | 91 |
| 11/17/2016 | Cobble | 27 | 97 |
| 11/21/2016 | Cobble | 30 | 72 |
| 11/21/2016 | Cobble | 75 | 29 |
| 11/21/2016 | Cobble | 11 | 49 |
| 11/21/2016 | Cobble | 10 | 19 |
| 11/21/2016 | Cobble | 55 | 61 |
| 11/21/2016 | Cobble | 14 | 72 |
| 11/30/2016 | Smooth | 20 | 59 |
| 11/30/2016 | Smooth | 40 | 69 |
| 11/30/2016 | Smooth | 10 | 25 |
| 11/30/2016 | Smooth | 10 | 66 |
| 11/30/2016 | Smooth | 24 | 117 |
| 11/30/2016 | Smooth | 20 | 35 |
| 11/30/2016 | Smooth | 40 | 13 |

used method for extrapolating the unmeasured upper and lower portions of a velocity profile when processing acoustic Doppler current profiler (ADCP) measurements [19]. The depth-averaged velocity was calculated by taking an average of the velocities of the profile weighted by the depth interval of each measurement.

The velocity profiles also enabled calculation of velocity ratios (surface velocity to depth-averaged velocity) for each set of conditions. Researchers of non-contact streamflow measurement methods such as LSPIV and SVR have placed recent focus on the velocity ratio as a critical measure of the stream's velocity distribution [18] [20]. The practical use of these methods relies upon selecting accurate estimates of velocity ratios with taking any additional velocity measurements below the water surface. One promising approach is to develop empirical relationships between the velocity ratio and easily quantifiable characteristics of a stream, such as substrate size and water depth. While the development of empirical relationships that may be applied outside of the laboratory setting in which our data was collected was not the objective of this study, the topic is critical to the surface float and velocity head rod methods. As a result, the relationship between depth, substrate, and velocity ratio was investigated and presented here.

**Measurement precision and accuracy.** The scaled mean error (SME) was used to calculate bias in the measurements. The SME measures how far the alternative-method measurements are from the ADV and PIV depth-averaged velocity. The SME takes on a positive value

if the measurement is an over-estimate of the control velocity and a negative value if it is an under-estimate. The formula for SME is:

$$SME = (V_{measured} - V_{control})/V_{control} \qquad (5)$$

Where, $V_{measured}$ is the velocity measured by the alternative method being tested and $V_{control}$ is the velocity measured with the ADV (water column profile) and PIV (surface). The coefficient of variation (CV) was used to calculate measurement precision. Unlike the SME, the CV is calculated independently of the measurements made by the ADV and PIV.

**Velocity head rod calibration and sampling interval.** It has been previously recognized that the theoretical equivalence of the kinetic energy of free-flowing water and the potential energy of water stagnating at the velocity head rod is not exact and that a calibration to known velocities is necessary [11]. Two hypothesized reasons for this are that kinetic energy is dissipated in other forms and that the head may be more representative of the average velocity of the upper portion of the stream than the depth-averaged velocity [11].

The method used by Fonstad et al. [11] to calibrate the velocity head rod measurements was to first apply the theoretical relationship to the measured heads to acquire uncalibrated velocities, then to compare these uncalibrated velocities to known depth-averaged velocities, and finally to derive a linear relationship between the uncalibrated velocities and the depth-averaged velocities. The same approach was used for this study with one addition. The velocity profiles were divided into several intervals to determine whether the velocity head is most affected by the depth-averaged velocity or the velocity in another depth interval. The uncalibrated velocities were compared to the depth-averaged velocities, surface velocities, and the average velocities in the top 5 cm, 10 cm, and 15 cm of the water column.

**Rising body vertical acceleration.** The rising body method employs the simple formula of downstream distance divided by travel time to calculate the depth-averaged velocity. This formula assumes that the body maintains a constant vertical velocity as it rises. A buoyant object will experience a period of acceleration, however, until it approaches its terminal velocity. Additionally, this acceleration will vary depending upon the net buoyancy and the shape of the object. Vertical acceleration may introduce a challenging bias to the rising body method. A relatively slow acceleration of the object to its terminal velocity would produce a bias toward underestimating the depth-averaged water velocity because the object spends more time at greater depths where velocities are typically lower. Slower accelerations generate greater bias (i.e., larger underestimations). Correcting for this bias universally would be challenging as it varies with object shape, object buoyancy, and stream depth.

Three side-by-side sets of measurements were made using two floats filled with different volumes of water to test whether the buoyancy of an object affects the measured depth-averaged velocity. The measurements were made using three different sets of conditions: 1) cobble substrate, 75 cm water depth and 29 cm/s depth-averaged velocity; 2) cobble substrate, 30 cm water depth and 72 cm/s surface velocity; and 3) smooth substrate, 40 cm water depth and 69 cm/s surface velocity. For each set of conditions, repeat measurements were taken by alternating between the two floats and a t-test was applied to the paired velocity measurements.

The rising body data were also examined for the presence and impact of vertical acceleration by comparing the average vertical velocity with the water depth for each fill volume. The vertical velocity was calculated by dividing the water depth by the rise time and the velocity residual was calculated by subtracting the ADV- and PIV-derived depth-averaged velocity from the rising body velocity. Pearson correlation statistics were calculated between water depth and vertical velocity.

**Rising air bubble vertical velocity.** The vertical velocity of rising air bubbles was calculated as $V_v$ according to a rearrangement of Eq (4):

$$V_v = V * d/d_d \qquad (6)$$

These values were calculated using the measured quantities of water depth (d) and downstream distance ($d_d$) the depth-averaged water velocity ($V_v$) as measured by the ADV (water column) and PIV (surface). The acceleration and terminal velocity of rising air bubbles are impacted similarly as those of rising floats in the rising body method, except that the impact is more complex because bubbles are not rigid bodies. No experiments were made to assess differences between accelerations and terminal velocities for air bubbles and the rigid floats used to test the rising-body method.

## Results

### Velocity profiles

The magnitude of the velocity ratio was dependent on substrate roughness (clast size) and water depth, with shallower depths and rougher substrates yielding smaller velocity ratios. This finding is consistent with previous studies [18]. All velocity profiles collected with the ADV and PIV were normalized and are shown together in Fig 1. For each profile, measurement depth was divided by water depth and the result subtracted from 1 to obtain the normalized measurement height above the bottom, and velocity was divided by depth-averaged velocity. The opacity of points is proportional to total water depth, with increasing opacity corresponding to increasing water depth. Velocity profiles have greater uniformity when the bed is smoother and water depths are greater. While the velocity profiles have different shapes for different substrates, their depth-averaged velocities commonly occur near a relative depth of 0.6 (a relative height of 0.4 in Fig 1). This is consistent with the theory and observations of flow in open channels [21]. At any given relative depth, the variance of normalized velocities is greater for rougher substrates than smoother substrates. This means that in channels with rougher substrates, such as in small and rocky montane streams, representing an entire velocity profile with a single measurement introduces significant measurement uncertainty. This increasing measurement uncertainty with increasing roughness is also illustrated in Fig 2, which shows histograms of velocity ratios for each substrate.

Velocity ratios are plotted versus water depth in Fig 3. For the gravel and cobble substrates, a nearly linear relationship between velocity ratio and water depth is apparent, but for the smooth substrate the velocity ratio does not appear to vary in a similar manner with water depth. For the gravel and cobble substrates, the relationship can be expressed as

$$\alpha = 0.00628 \, (1/cm) * d + 0.465 \qquad (7)$$

Where, $\alpha$ is the velocity ratio and $d$ is the water depth (in cm; ranging from 10 to 75 cm). This relationship has a correlation coefficient of 0.95 and an RMSE of 0.038. A linear relationship between velocity ratio and depth was explored for the smooth substrate, with a correlation coefficient of 0.69 and a RMSE of 0.033. However, because of the fewer number of samples, the narrow domain of depth values tested, the lower correlation coefficient, and because of how close all velocity ratios values were to the approximate upper limit of 1, it was considered more appropriate to set the velocity ratio as a constant value equal to the average velocity ratio of 0.92. The RMSE of this approach is 0.050. For the alternative methods that require a velocity ratio (the surface float and velocity head rod methods), these relationships were used to estimate the velocity ratios. This approach was favored over using the exact velocity ratio for each set of conditions to better mimic how unknown velocity ratios are approximated in the field. It

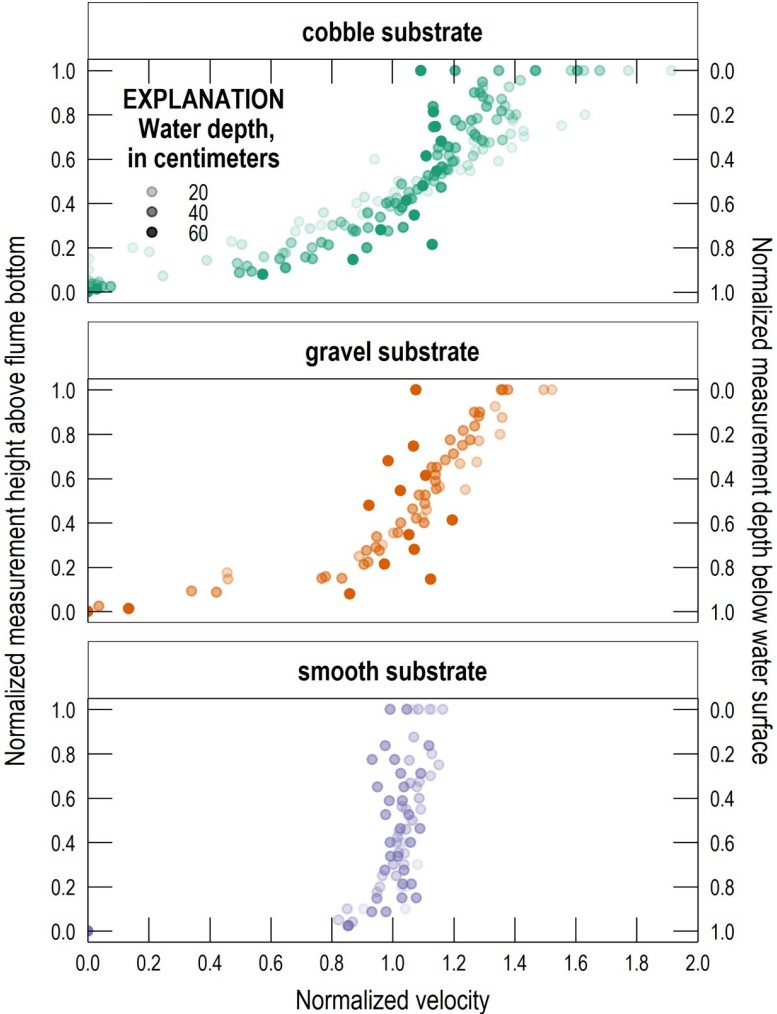

**Fig 1. Vertical velocity profiles for each set of conditions tested in the flume.** Measurement depth is normalized by water depth and the result subtracted from 1 to obtain normalized measurement height above the bottom, and velocity is normalized by depth-averaged velocity. The point opacity is proportional to water depth, with increasing opacity corresponding to increasing water depth.

should not be assumed that these relationships apply generally to all streams or to water depths outside the range tested in this study.

## Surface float method

**Precision and accuracy.** Summary performance metrics for the surface float and other alternative methods are listed in Table 2. The precision of the surface float method is comparable to the other methods, but the method's bias is greater with an SME of -0.08. The surface float method tends to underestimate the depth-averaged velocity.

The CV and SME of the surface float and other alternative methods were compared to water depth and depth-averaged velocity to see how precision and bias vary with these variables (Fig 4). A table of Pearson correlation coefficients and associated p-values are listed in Table 3. For the surface float method, a positive relationship was observed at the 95% confidence level between SME and water depth. The surface float method's bias towards underprediction is greater at smaller depth-averaged velocities (Fig 4).

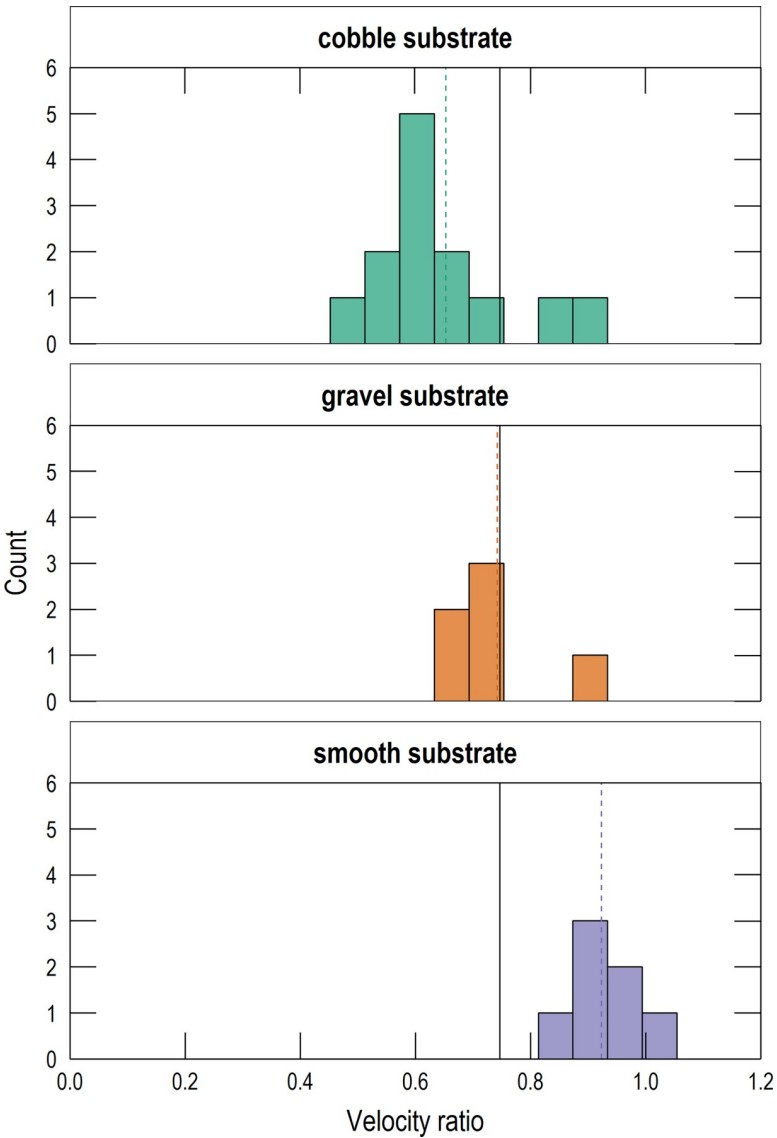

**Fig 2. Histograms of velocity ratios for each substrate.** The dashed vertical lines are the mean velocity ratio for each substrate. The solid black vertical line is the overall mean velocity ratio.

**Practical considerations.** No practical challenges were encountered in performing surface float measurements in the laboratory setting. Under natural conditions, turbid water or wide streams may make it impractical to place a reference object on the stream bed or bank. Overall, the surface float method is the easiest to perform.

The surface float method has two main drawbacks. First, it appears to have an under-estimation bias, particularly at smaller velocities. Second, the method relies upon the use of a velocity ratio, which may be difficult to estimate for shallow streams with rough substrates.

## Velocity head rod method

**Calibration and sampling intervals.** The velocity head rod method measured velocities greater than 15 cm/s, below which the head was not observable. At velocities greater than 40–

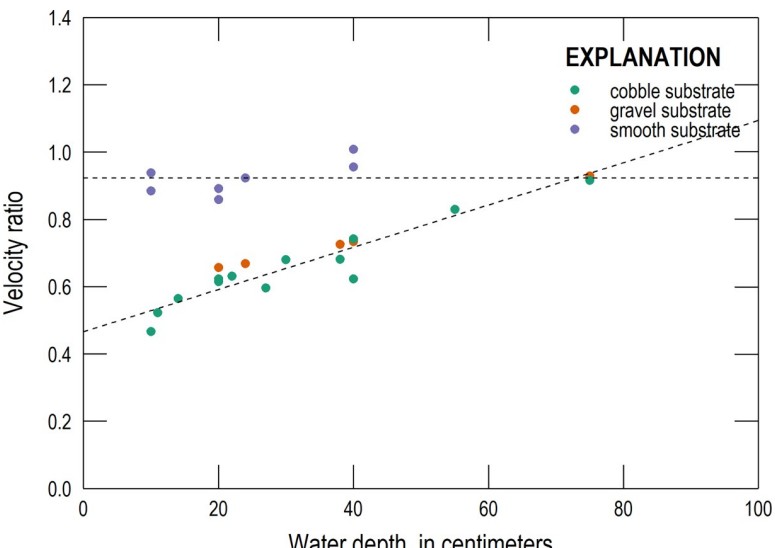

**Fig 3. Velocity ratio versus water depth for all conditions.** The horizontal dashed line shows the average velocity ratio of 0.92 for the smooth substrate; the second dashed line is the line of best fit between velocity ratio and water depth for the gravel and cobble substrates.

50 cm/s, waves formed on the stream surface and the precision of the head measurement decreased.

We found that the measured velocities (uncalibrated velocities) correlated most closely (largest $R^2$) with average velocities measured in the top 5 cm of the velocity profile. Despite these findings, it would be easier in practice to use a calibration formula that either translates the measured velocity to a depth-averaged velocity or to a surface velocity from which a depth-averaged velocity can be computed using an appropriate velocity ratio. Scatter plots and regression lines are shown in Fig 5 for the measured (uncalibrated) velocities versus the surface velocity (additional figures are included as supplemental material). These and several other approaches were compared with a 5-fold cross validation. A cross validation is a way to test how a model, in this case a linear regression model, may perform when used to make predictions for values of input variables that were not included in set used to develop the model. A cross-validation tests this by removing a random subset of data from the overall dataset, fitting a model to this reduced set, and then testing how well the model can predict the values of the set that was withheld. The error in these predictions are called the out-of-set error. For a 5-fold dataset, this is done 5 times, with a different subset comprising 1/5 of the overall dataset withheld each time. We found that calibrating the measured velocity to the surface velocity (and then multiplying by a velocity ratio) had the lowest out-of-set error. (Results of the different approaches are provided in the supplemental material.) The final formula for the velocity head rod method is thus:

$$V_s = 1.098 * \left( \sqrt{2gh} - 15.31 \frac{cm}{sec} \right) \tag{8}$$

**Table 2. Summary of precision and bias for each alternative method.**

|  | Surface float method | Velocity head rod method | Rising body method | Rising air bubble method |
|---|---|---|---|---|
| Mean CV | 0.10 | 0.10 | 0.14 | 0.09 |
| Mean SME | -0.08 | -0.04 | -0.01 | 0.04 |

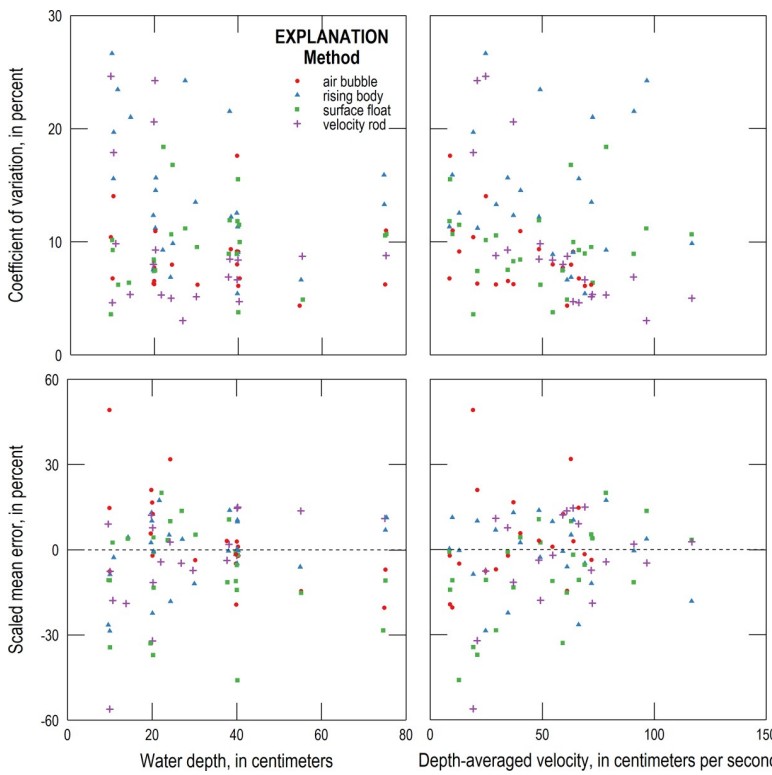

**Fig 4. COV and SME versus water depth and depth-averaged velocity for each alternative velocity method.**

Where, $V_s$ is the surface velocity in cm/sec, $g$ is gravitational acceleration in cm/sec$^2$, and $h$ is the head difference in cm. This formula applies to all substrate types. As with the surface float method, the depth-averaged velocity can be calculated by multiplying the surface velocity by a velocity ratio.

**Precision and accuracy.** Overall, the velocity head rod method has moderate precision and accuracy compared to the other three alternative methods, with a mean CV of 0.1 and mean SME of -0.04 (Table 2). For the velocity head rod method, two relationships between measurement performance metrics and stream variables were observed at the 95% confidence level. First, precision (CV) was found to strongly improve with depth-averaged velocity, as can be seen clearly in Fig 4. This seems counter to the challenge users had measuring a precise head value with the oscillations that occurred at depth-averaged velocities greater than 40–50

**Table 3. Correlations between the performance metrics (CV and SME) and the stream variables (water depth and depth-averaged velocity) for each alternative velocity method.**

| Method | Variable | Performance Metric | Correlation coefficient | p-value |
|---|---|---|---|---|
| velocity head rod | depth-averaged velocity | CV | -0.76 | 0.000 |
| air bubble | water depth | SME | -0.64 | 0.002 |
| surface float | depth-averaged velocity | SME | 0.53 | 0.005 |
| velocity head rod | water depth | SME | 0.53 | 0.017 |
| air bubble | depth-averaged velocity | CV | -0.49 | 0.028 |
| rising body | water depth | SME | 0.39 | 0.057 |
| velocity head rod | depth-averaged velocity | SME | 0.40 | 0.077 |

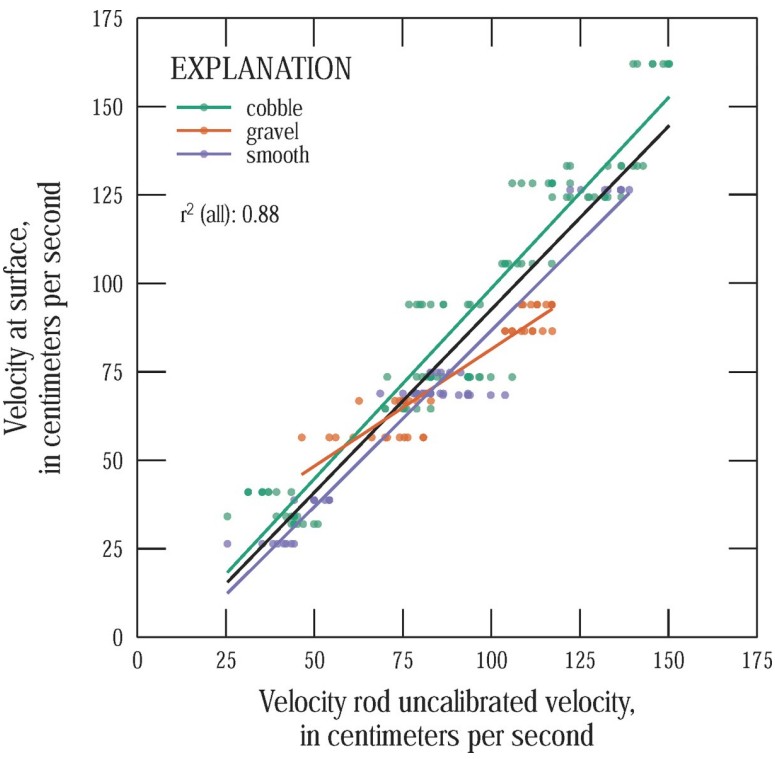

**Fig 5. Surface velocity versus (uncalibrated) velocity measured with the velocity head rod method.** The colored lines are best fit lines for each substrate type and the black line is the best-fit line for all measurements.

cm/s. The theoretical equivalence of kinetic and potential energy that underpins this method, however, predicts that the head will increase with the square of velocity, suggesting that the sensitivity (precision) of the technique should improve as velocity increases. A positive correlation between SME and water depth is also observed at the 95% confidence level. At the 90% confidence level, a positive relationship was found between depth-averaged velocity and SME. These latter two relationships appear to be driven largely by underpredictions of depth-averaged velocity at smaller depths and velocities.

**Practical considerations.** Measurements with the velocity head rod were generally fast and simple to perform. Positioning the observer's eyes at the level of the water surface is challenging at shallow depths, however, if the user is unwilling or unable to kneel in the stream. As previous practitioners have found, a clear velocity head rod improves the ability to measure the head [11]. While the head was found to oscillate at greater depth-averaged velocities and reduce the precision of the head measurement, the exponential relationship between velocity and head counteracts this and leads to improved precision of head measurements as the velocity increases.

The velocity head rod method has two main drawbacks. First, the method's precision declines with smaller velocities and the method does not work for depth-averaged velocities smaller than 15 cm/s. Second, like the surface float method, the velocity head rod method requires a velocity ratio (Eq 6), which may be difficult to estimate in a field setting.

## Rising body method

**Vertical acceleration.** Tests were performed to measure how the period of acceleration in the rising body's vertical velocity affect the bias of the depth-averaged velocity determined

**Table 4. T-test results for paired sets of rising body measurements.**

| Substrate | Water Depth (cm) | Depth-averaged velocity (cm/s) | Fill volume for float 1 (ml) | Fill volume for float 2 (ml) | p-value |
|---|---|---|---|---|---|
| Cobble | 75 | 29 | 4.36 | 9.375 | 0.65 |
| Cobble | 30 | 72 | 5 | 9.375 | 0.33 |
| Smooth | 40 | 69 | 6.25 | 9.375 | 0.79 |

with the method. Depth-averaged velocities were measured for different combinations of substrates and water depths using two floats side by side (Table 4). For each set of measurements, the floats were filled with different amounts of water (fill volumes). These measurements of varying substrate types and water depths indicate no significant differences in the measured depth-averaged velocities for two different fill volumes of the rising body (Table 4).

While the side-by-side tests revealed no significant difference between depth-averaged velocities measured with floats of different buoyancies, there is a degree of vertical acceleration that can be discerned from the full set of rising body measurements. For each individual measurement, the mean vertical velocity of the float can be calculated by dividing the water depth by the time taken for the body to rise to the surface. This mean vertical velocity, which is made up of lower vertical velocities as the float begins its ascent at the stream bottom and high vertical velocities as the float approaches the water surface, should increase with increasing depth. This was tested by dividing the rising body measurement into six groups based upon the fill volume of the float and for each group running a Pearson's correlation test for mean vertical velocity and water depth. The results, shown in Table 5, show that for 4 of the 6 fill volumes a significant positive correlation between mean vertical velocity and depth, suggesting that a noticeable degree of acceleration occurred during the rising body measurements. The reason the impact of this acceleration was not observed in the side-by-side tests may have been due to additional measurement error present when performing those tests, which involved measuring both time and distance travelled for two different floats.

**Precision and accuracy.** Overall, the rising body method has the lowest precision of the four alternative methods tested, but it also has the least bias, with a mean CV of 0.14 and mean SME of -0.01. No relationships were observed between performance metrics and stream variables at the 95% confidence level. At the 90% confidence level, a positive relationship was observed between SME and water depth. This relationship may be the result of the observed vertical acceleration of the rising body although the side-by-side measurements cast doubt upon this conclusion.

**Practical considerations.** As hypothesized, the rising body method showed greater accuracy than the other alternative methods. This is likely because the method directly measures the depth-averaged velocity and does not require a velocity ratio adjustment.

A few practical limitations were encountered in applying the rising body method. Most importantly, it was challenging to make measurements using the method with fewer than

**Table 5. Correlation coefficients of water depth and vertical velocity for rising floats filled with different volumes of water.**

| Fill volume (ml) | Correlation coefficient | Number of samples | P-value |
|---|---|---|---|
| 5 | 0.6 | 4 | 0.00 |
| 7.5 | 0.34 | 4 | 0.01 |
| 8.125 | 0.71 | 2 | 0.00 |
| 8.75 | 0.25 | 3 | 0.19 |
| 9.375 | 0.64 | 7 | 0.00 |
| 9.5 | 0.01 | 6 | 0.92 |

three people. The method would be easier to undertake with a float that is disposable, which may allow two people to make a measurement. In addition, it was challenging to hold and release the float without noticeably disturbing the direction of flow. Field conditions may introduce additional challenges. While the float appeared to accelerate vertically, it is difficult to determine whether and to what extent this caused a bias at smaller depths.

### Rising air bubble method

**Vertical air bubble velocity.** A histogram of the calculated vertical air bubble velocities is shown in Fig 6. The mean vertical velocity is 19.4 cm/s and the standard deviation is 3.5 cm/s. This relatively large variance suggests that it may be too simplistic to use a single vertical velocity. If the true vertical air bubble velocity were less than the mean value, the horizontal velocity ($V$ in Eq 2) would be overestimated if the mean vertical velocity is used for $V_v$ in Eq 2. The opposite would happen if the true vertical air bubble velocity were greater than the mean value.

**Precision and accuracy.** The rising air bubble method had a mean CV of 0.09 and a mean SME of 0.04, suggesting that the method had the highest precision of the four alternative methods and a bias comparable to that of the velocity head rod method. Measurements with the air bubble method, however, were too difficult to make for six of the 26 conditions tested and thus the overall performance metrics of the method could not be compared to those of the other three methods. For the rising air bubble method, two relationships were observed at the 95% confidence level between performance metrics and stream variables. First, bias had a strong negative correlation with water depth. Bias was positive at shallow depths (0–30 cm), neutral at intermediate depths (30–50 cm), and negative at large depths (50–80 cm). This bias trend likely results from assuming a single vertical air bubble velocity. The precision of the air bubble method improved with increased stream velocity. Unfortunately, it was at greater stream velocities (greater than ~70 cm/s) that measurements were frequently too difficult to make.

**Practical considerations.** In most cases, the rising air bubble method was easy to carry out by one person. For the conditions where measurements were possible, the method also showed high precision. Even though the method requires an independent estimate of the

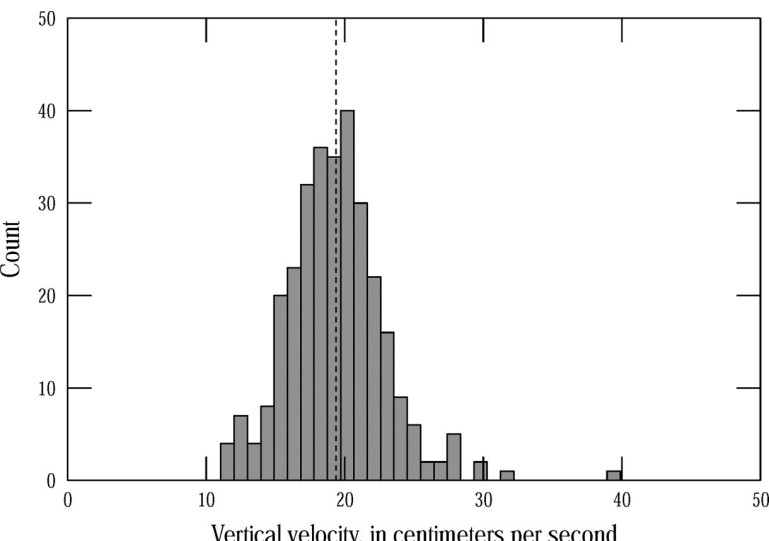

**Fig 6. Histogram of calculated vertical air bubble velocities.** The vertical dashed line shows the mean vertical velocity of 19.4 cm/s.

vertical air bubble velocity, and thus does not directly measure the depth-averaged velocity in the same manner as the rising body method, the rising air bubble method results in a measurement that represents the entire water column. For this reason, measurements made with the rising air bubble method do not require a velocity ratio adjustment.

Numerous practical challenges were encountered with the rising air bubble method. It was challenging to see the air bubbles and identify where they surfaced, particularly at large stream velocities (greater than 40–50 cm/s) when waves were present on the water surface. Visibility may be an even greater challenge in natural streams. Several factors also appeared to affect the size of the air bubbles and their resultant buoyancy. Water depth appeared to be the dominant factor (greater hydrostatic pressure resulted in smaller and less buoyant bubbles), but the position of the nozzle from which air bubbles originated and the presence of turbulence near the streambed also appeared to have an impact. The most important challenge of the rising air bubble method, however, were the numerous materials and complicated assembly the method required. It may be too difficult to construct a sufficiently standardized device among a disparate population of users to produce comparable measurements. For this reason, we do not consider the method suitable for our intended application. Under different settings, however, the method may show promise, and avenues exist for overcoming the challenges experienced in this experiment. For example, researchers are using digital instrumentation [13] and cameras with image processing algorithms [22] to enhance the utility of the rising air bubble method.

## Comparative performance

The relative precision, as measured by the CV, of the four alternative methods under different conditions is shown in Fig 7. The absolute value of the SME under different conditions is shown in Fig 8. In each figure, differently colored circles represent different measurement techniques, and the size of each circle represents the relative CV or SME value, with larger circles representing greater values (less precision or more bias) and smaller circles representing

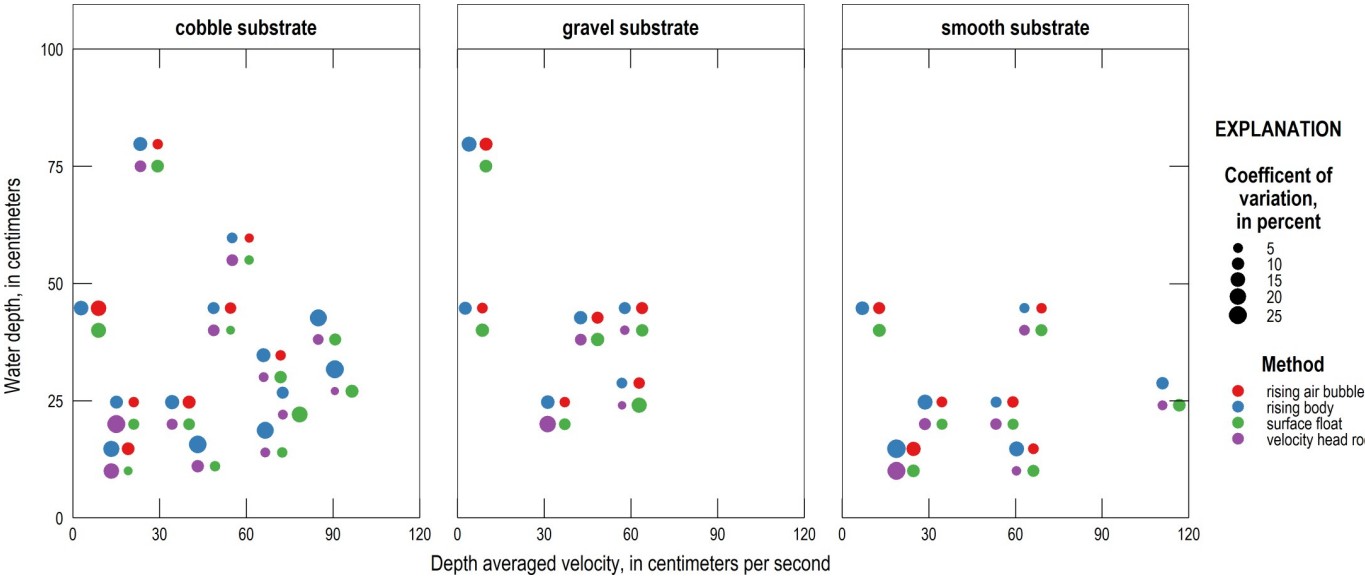

**Fig 7. Relative precision, as measured by the CV, of the four alternative velocity methods under different conditions.** Different colored circles represent different measurement techniques and the size of the circle represents the relative CV value, with larger circles showing less precision and smaller dots more. Each group of colored bubbles is centered on the coordinates representing the tested water depth and depth-averaged velocity. Individual bubbles are offset from this position, however, to avoid overlap.

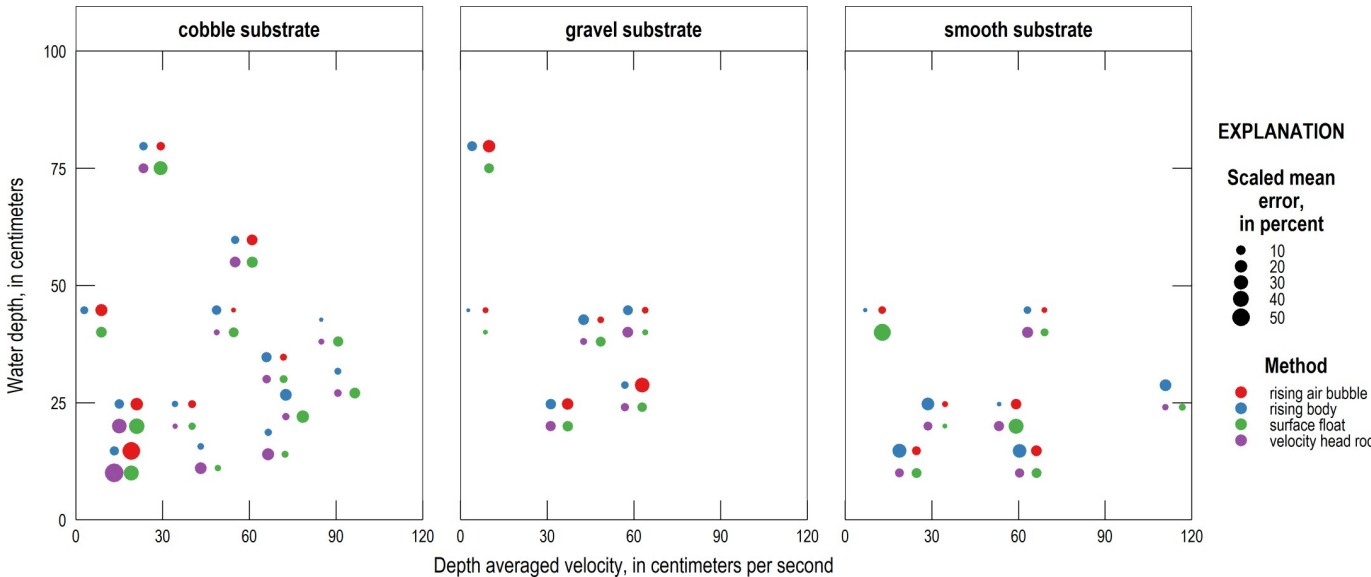

**Fig 8. Accuracy, as measured by the SME, of the four alternative velocity methods under different conditions.** Each color circle represents a different measurement technique and the size of the bubble represents the relative SME value, with larger circles showing less accuracy and smaller dots more. Each group of colored bubbles is centered at the coordinates representing the depth and velocity tested, however the individual bubbles are offset from this position to avoid overlap.

smaller values (more precision or less bias). The results highlight that the substrate appears to be a major factor in determining measurement accuracy but only a modest factor in determining measurement precision.

It is important to note that to calculate SME, an estimate of depth-averaged velocity had to be obtained for each of the alternative methods. In the case of the surface float and velocity head rod, this involves applying the velocity ratio as a multiplier. The specific velocity ratio used for each measurement was found using a relationship between the velocity ratio, as solved using ADV and PIV measurements, and the total water depth. Because the relationship between velocity ratio and depth was derived as a best fit line between the known quantities, the measurements of surface float and velocity head rod that use this best-fit velocity ratio can be considered to have a degree of bias correction applied prior to the calculation of the SME. Importantly, this may underestimate the bias compared to a measurement taken in field conditions for which the velocity ratio is harder to estimate. The CV, on the other hand, is insensitive to scale and is thus not affected by the choice of velocity ratio.

## Discussion

The primary goal of this study was to estimate the precision and accuracy of four stream velocity measurement methods. A secondary goal of this study was to develop standardized instrumentation and methodology. In both cases, the findings of this study are judged against the instruments and methods that are traditionally used for measuring streamflow—in particular, wading measurements collected with mechanical or acoustic current velocity meters. The ease and affordability of the proposed alternative methods may come at the cost of lower data quality (precision and accuracy) relative to traditional stream gaging. With this study, we hope to explore this tradeoff between measurement ease and quality and, if possible, to reduce it.

For the purposes of developing methodologies of the alternative stream velocity methods and in judging their performance against tradition methods, it will be important to identify

whether there are certain stream conditions under which one of the four alternative methods performs best. The computed precision and bias and observed practical constraints can help answer this question. A major choice to be considered is whether to use the rising body method, with its small bias but lesser precision, or the surface float or velocity head rod methods that have greater bias but greater precision. The intended application of the velocity measurements and the number of people available to help make them will guide this choice. If large bias would be detrimental to an application and measurements must be made in shallow and rocky (rough substrate) streams where the surface methods (i.e., surface float or velocity head rod methods) may produce bias, then it may be worth taking the extra effort to use the rising body method. Multiple measurements could be made with the rising body method to improve the depth-averaged velocity estimate. In other cases, when the velocity head rod or the surface float methods would be suitable, the stream velocity itself can help determine the better choice of method. For example, at stream velocities greater than ~60 cm/s the velocity head rod method has greater precision and at smaller stream velocities the surface float method has greater precision. This tradeoff may allow for these two methods to complement one another. It may be advantageous to use the velocity head rod method to measure faster currents near the middle of a stream and the surface float method to measure slower currents near the stream banks.

Ultimately, these alternative stream velocity measurement methods will be judged against their traditional counterparts, such as mechanical or acoustic flow meters or modern non-contact methods. Even though methods that use traditional instruments face challenges in shallow streams with rough substrates, few studies have characterized the accuracy of traditional methods in such streams. One study of stream-gaging techniques in two small, slow streams found SME values of ~0.2–0.25 for traditional methods using a Marsh McBirney meter, ~0.35–0.6 for a Price Type-AA meter, and ~0.2–0.7 for a Pygmy meter [23]. In the same streams, SME values of ~0.35 for the more sophisticated version of the rising body method [24] (one-orange method) compared favorably to these traditional velocity measurement methods. While the SME values were obtained by comparing stream discharge and not depth-averaged velocity, they do suggest that non-traditional velocity measurement methods may perform as well as traditional methods the farther conditions in a small stream deviate from ideal measurement conditions.

Ultimately, the major advantage of these alternative methods is their low cost, portability of equipment, and easy maintenance. Backcountry rangers, canal operators, or citizen scientists can easily carry and use these instruments in remote locations. With widespread adoption, using these velocity measurement techniques with the velocity-area method of measuring discharge could generate a valuable dataset of stream discharge measurements where few measurements currently exist. The recent developments in online repositories [25] [26] [27] and digital communication of citizen science data suggest that an opportunity exists for non-traditional velocity measurement methods to increase the body of knowledge in the fields of hydrology and water resources. A collaborative effort of many parties, with measurements stored in an accessible digital repository, could result in exponential growth of information where individual or spatially limited observations currently provide only cursory information of watershed system function.

## Conclusions

A laboratory experiment was conducted to test four current velocity measurement methods that can be used to quantify streamflow in small streams. Through repeated measurements under controlled and instrumented conditions, the precision and bias of these methods were determined. The surface float and velocity head rod methods were both found to be easy to

perform with relatively great precision; with the surface float method performing better at depth-averaged velocities smaller than ~60 cm/s and the velocity head rod method performing better at depth-averaged velocities greater than ~60 cm/s. These methods measured the surface, or near surface, velocity rather than the depth-averaged velocity and required adjustment using an appropriate velocity ratio to compute depth-averaged velocity. The observed strong relationship between stream depth and velocity ratio that was consistent for gravel and cobble substrates suggests that an empirical relationship could be developed for estimating velocity ratios. The rising body method had the least precision but also the least bias. The method was more cumbersome to execute, requiring more personnel. Finally, the rising air bubble method was considered unsuitable for the type of field applications anticipated because of factors that could impact air bubble creation, movement, and visibility. The three methods deemed suitable–surface float, velocity head rod, and rising body—all have promise as tools for collecting fast and low-cost streamflow measurements on small streams that are typically omitted from streamflow monitoring networks.

## Supporting information

**S1 Table. Table of cross-validation results for different strategies of calibrating the velocity head rod.**
(DOCX)

**S1 Fig. Diagram of the four stream velocity measurement methods tested in this study.** Subfigure A shows the surface float method, with the required materials of a water-fillable float, yardstick, and stopwatch. Subfigure B shows the rising body method, with the required materials of a water-fillable float, yardstick, and stopwatch. Subfigure C shows the velocity-head rod method, with the required materials of a velocity head rod and a ruler. Subfigure D shows the rising air bubble method, with the required materials of a bubbler and a yardstick.
(TIFF)

**S2 Fig. Vertical velocity profiles for the 6 conditions without a usable PIV measurement of surface velocity.** The blue dots show the velocity measured by the flume ADV. Different shades of blue are profiles measured at different times. A power law curve fit to the ADV measured velocities is shown as a black curve and the extrapolated surface velocity is shown as an unfilled black circle.
(TIFF)

**S3 Fig. Depth-averaged velocity versus (uncalibrated) velocity measured with the velocity head rod method.** The colored lines are best fit lines for each substrate type and the black line is the best-fit line for all measurements.
(TIFF)

**S4 Fig. Velocity of upper 5 cm versus (uncalibrated) velocity measured with the velocity head rod method.** The colored lines are best fit lines for each substrate type and the black line is the best-fit line for all measurements.
(TIFF)

**S5 Fig. Velocity of upper 15 cm versus (uncalibrated) velocity measured with the velocity head rod method.** The colored lines are best fit lines for each substrate type and the black line is the best-fit line for all measurements.
(TIFF)

## Acknowledgments

The authors thank the many thoughtful reviewers who contributed to the content of this paper. Bob Basham, Ralph Budwig, and Jenna Duffin of the University of Idaho provided invaluable assistance with flume operations, measurement collection, and planning. Ron Schulz, Nancy Youngblood, Crystal Sverdsten, and Michael Walrath of USGS all generously volunteered four long days of their time to collect measurements.

Any use of trade, firm, or product name is for descriptive purposes only and does not imply endorsement by the U.S. Government.

## Author Contributions

**Conceptualization:** Stephen Hundt, Kyle Blasch.

**Data curation:** Stephen Hundt.

**Formal analysis:** Stephen Hundt.

**Funding acquisition:** Kyle Blasch.

**Investigation:** Stephen Hundt, Kyle Blasch.

**Methodology:** Stephen Hundt, Kyle Blasch.

**Project administration:** Stephen Hundt, Kyle Blasch.

**Resources:** Stephen Hundt.

**Supervision:** Kyle Blasch.

**Visualization:** Stephen Hundt.

**Writing – original draft:** Stephen Hundt.

**Writing – review & editing:** Kyle Blasch.

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
