## [Decision Letter · Decision Letter 0]

24 Jul 2019

PONE-D-19-17424

Laboratory assessment of alternative stream velocity measurement methods

PLOS ONE

Dear Mr. Hundt,

Thank you for submitting your manuscript to PLOS ONE. After careful consideration, we feel that it has merit but does not fully meet PLOS ONE’s publication criteria as it currently stands. Therefore, we invite you to submit a revised version of the manuscript that addresses the points raised during the review process.

We would appreciate receiving your revised manuscript by Sep 07 2019 11:59PM. To enhance the reproducibility of your results, we recommend that if applicable you deposit your laboratory protocols in protocols.io, where a protocol can be assigned its own identifier (DOI) such that it can be cited independently in the future. For instructions see: http://journals.plos.org/plosone/s/submission-guidelines#loc-laboratory-protocols

We look forward to receiving your revised manuscript.

Kind regards,

Vassilis G. Aschonitis

Academic Editor

PLOS ONE

Journal Requirements:

1. We note that you have stated that you will provide repository information for your data at acceptance. Should your manuscript be accepted for publication, we will hold it until you provide the relevant accession numbers or DOIs necessary to access your data. If you wish to make changes to your Data Availability statement, please describe these changes in your cover letter and we will update your Data Availability statement to reflect the information you provide.

2. We note you have included a table to which you do not refer in the text of your manuscript. Please ensure that you refer to Tables 1,3, & 5 in your text; if accepted, production will need this reference to link the reader to the Table.

Reviewers' comments:

Reviewer's Responses to Questions

**Comments to the Author**

1. Is the manuscript technically sound, and do the data support the conclusions?

Reviewer #1: Yes

2. Has the statistical analysis been performed appropriately and rigorously? 

Reviewer #1: Yes

3. Have the authors made all data underlying the findings in their manuscript fully available?

Reviewer #1: Yes

4. Is the manuscript presented in an intelligible fashion and written in standard English?

Reviewer #1: Yes

5. Review Comments to the Author

Reviewer #1: General comment

This study evaluates the efficiency and accuracy of four alternative, low-cost, methods to measure the stream velocity under laboratory conditions. The paper is well structured and clearly written, providing all necessary information in the various sections (introduction, methodology, results etc.) in an appropriate and diligent way. In my opinion, the paper may be published after minor revision. Here are my specific comments.

Specific comments

Line 30: Please delete “with the velocity head rod and surface float methods”

Lines 72-79: This paragraph is not appropriate for the Material and Methods Section. Please incorporate the meaning of this paragraph in the Discussion or Conclusion section.

Line 154: Please explain how this formula is derived. Based on Bernoulli equation?

Line 154: Please explain how this formula is derived. Based on Bernoulli equation?

Line 190: Please explain or provide a reference how this formula is derived.

Line 201 (also lines 353, 359, 456): Please note that a broken link is appeared (Error! Reference source not found).

Lines 239-241: The meaning in this sentence is not clear. Please consider to revise more clearly.

Fig. 1: Does the same title should be appeared in both vertical axes? If yes, why are the figures upside down?

Line 337: What is the correlation coefficient for equation 5? Please provide.

Lines 339-340: For the smooth substrate, a constant velocity ratio is proposed, obviously for convenience in use. In my opinion, a new formula, similar to equation 5, would describe more accurately the experimental data and should be provided. In every case, the correlation coefficient should be given even for the case of constant velocity ratio (a =0.92) as well as for the new formula.

Fig. 4: Please correct the division in the horizontal axes of the upper two diagrams to comply with the horizontal axes of the two lower diagrams.

Line 400: How is the velocity ratio estimated in equation 6? Which values of velocity ratio should be used? Please clarify. Also, is it possible to use equation 6 to estimate the surface velocity? If not, please clarify why.

Line 620: Which are the three methods? Please clarify.

6. PLOS authors have the option to publish the peer review history of their article (what does this mean?). If published, this will include your full peer review and any attached files.

Reviewer #1: Yes: Charalampos Doulgeris

---

## [Author Response · Author response to Decision Letter 0]

12 Aug 2019

Dear reviewers,

Thank you for providing your review to the article entitled Laboratory Assessment of Alternative Stream Velocity Measurement Methods. The points raised were all helpful and relevant, and I am raising only one minor rebuttal. Otherwise, I addressed each point raised by making the correction or clarification that was suggested. Responses to each point are included below.

Sincerely,

Stephen Hundt

Editorial Review

1. We note that you have stated that you will provide repository information for your data at acceptance. Should your manuscript be accepted for publication, we will hold it until you provide the relevant accession numbers or DOIs necessary to access your data. If you wish to make changes to your Data Availability statement, please describe these changes in your cover letter and we will update your Data Availability statement to reflect the information you provide.

Noted

2. We note you have included a table to which you do not refer in the text of your manuscript. Please ensure that you refer to Tables 1,3, & 5 in your text; if accepted, production will need this reference to link the reader to the Table.

The Microsoft Word links were broken. I have removed those links and manually entered references to all tables.

Peer Reviewer #1

Reviewer #1: General comment

This study evaluates the efficiency and accuracy of four alternative, low-cost, methods to measure the stream velocity under laboratory conditions. The paper is well structured and clearly written, providing all necessary information in the various sections (introduction, methodology, results etc.) in an appropriate and diligent way. In my opinion, the paper may be published after minor revision. Here are my specific comments.

Thank you

Specific comments

Line 30: Please delete “with the velocity head rod and surface float methods”

Done

Lines 72-79: This paragraph is not appropriate for the Material and Methods Section. Please incorporate the meaning of this paragraph in the Discussion or Conclusion section.

Done

Line 154: Please explain how this formula is derived. Based on Bernoulli equation?

Yes. An explanation was provided.

Line 190: Please explain or provide a reference how this formula is derived.

An explanation of the formula’s derivation was provided.

Line 201 (also lines 353, 359, 456): Please note that a broken link is appeared (Error! Reference source not found).

Thank you. These have been fixed.

Lines 239-241: The meaning in this sentence is not clear. Please consider to revise more clearly.

I have revised with a clearer explanation.

Fig. 1: Does the same title should be appeared in both vertical axes? If yes, why are the figures upside down?

That was a mistake. The left y-axis has been corrected to ‘…height above…’ The dual y-axes and titles were the suggestion of a prior reviewer.

Line 337: What is the correlation coefficient for equation 5? Please provide.

I’ve included the correlation coefficient (0.95) and RMSE (0.038).

Lines 339-340: For the smooth substrate, a constant velocity ratio is proposed, obviously for convenience in use. In my opinion, a new formula, similar to equation 5, would describe more accurately the experimental data and should be provided. In every case, the correlation coefficient should be given even for the case of constant velocity ratio (a =0.92) as well as for the new formula.

I have included the correlation coefficient and RMSE for the smooth substrate depth vs velocity ratio. However, I still chose to apply a constant, and gave justification for this choice in the text. The correlation coefficient doesn’t make sense for this non-linear model, but I did include the RMSE (just the standard deviation in this case). 

Fig. 4: Please correct the division in the horizontal axes of the upper two diagrams to comply with the horizontal axes of the two lower diagrams.

Corrected.

Line 400: How is the velocity ratio estimated in equation 6? Which values of velocity ratio should be used? Please clarify. Also, is it possible to use equation 6 to estimate the surface velocity? If not, please clarify why.

This section was rewritten to add clarity.

Line 620: Which are the three methods? Please clarify.

This has been clarified.

---

## [Editor Report · Decision Letter 1]

27 Aug 2019

Laboratory assessment of alternative stream velocity measurement methods

PONE-D-19-17424R1

Dear Dr. Hundt,

We are pleased to inform you that your manuscript has been judged scientifically suitable for publication and will be formally accepted for publication once it complies with all outstanding technical requirements.

With kind regards,

Vassilis G. Aschonitis

Academic Editor

PLOS ONE

Additional Editor Comments (optional):

The authors have made all the appropriate revisions and the article can be accepted as is.
---

## [Editor Report · Acceptance letter]

29 Aug 2019

PONE-D-19-17424R1 

Laboratory assessment of alternative stream velocity measurement methods 

Dear Dr. Hundt:

I am pleased to inform you that your manuscript has been deemed suitable for publication in PLOS ONE. Congratulations! Your manuscript is now with our production department. 

With kind regards,

on behalf of

Dr. Vassilis G. Aschonitis 

Academic Editor

PLOS ONE